# Phosphorescent extensophores expose elastic nonuniformity in polymer networks

Kaikai Zheng ®[1], Yifan Zhang[1], Bo Li ®[1] & Steve Granick ®[1,2] ✉

Networks and gels are soft elastic solids of tremendous technological importance that consist of cross-linked polymers whose structure and connectivity at the molecular level are fundamentally nonuniform. Pre-failure local mechanical responses are not understood at the level of individual crosslinks, despite the enormous attention given to their macroscopic mechanical responses and to developing optical probes to detect their loci of mechanical failure. Here, introducing the extensophore concept to measure non-destructive forces using an optical probe with continuous force readout proportional to deformation, we show that the crosslinks in an elastic polymer network extend, fluctuate, and deform with a wide range of molecular individuality. Requiring little specialized equipment, this foundational single-molecule phosphorescence approach, applied here to polymer science and engineering, can be useful to a broad science and engineering community.

There are many ways to characterize chemical microenvironments and molecular structure of polymer networks[1] but to characterize mechanical microenvironments within them is an unsolved problem. Though mechanical failure can be detected by optical probes using bond breakage or isomerization[2–14], a robust single-molecule method to characterize pre-failure mechanical microenvironments is an unsolved problem. A single-molecule optical probe for this purpose should reversibly detect variable levels of force before failure. Here we report our experience using individual phosphorescent molecules at room temperature to detect variable levels of force before failure. We find this approach to be productive despite folklore in the single-molecule community that cryogenic temperature would be needed unless the background is exceptionally clear[15]. Figure 1a shows the chemical structure of the phosphorescent molecule we used. It is the same molecule developed by Filinenko and Khusnutdinova to study macroscopic mechanical deformation[6] except that we modified its synthesis by adding pendant carbon–carbon double bonds so it could be attached to polymer strands within elastic networks as a crosslink. Here, as proof-of-concept, we synthesize crosslinked polymethyacrylate (PMA) using UV-induced free radical polymerization. A parenthetical technical remark is that relative to alternative approaches, the extensophore concept introduced here avoids the photobleaching that is problematic when fluorescence is measured over extended times, and also avoids the problematical perturbative aspects of using nanoparticle optical probes. As we now describe, this enabled us to measure mechanical properties of individual crosslinks with direct relevance to unsolved fundamental questions of polymer science and engineering.

## Results

The microscope image in Fig. 1b shows what we interpret to be a single phosphorescent emitter in a crosslinked network of polymethacrylate. Averaged over 10 s to suppress background signal and minimize spurious quenching from molecular oxygen in the ambient air[16], the intensity profile is Gaussian with full-width-at-half maximum (FWHM) ~280 nm close to the diffraction limit, $d = 0.61 \Lambda/N_A$ ~ 300 nm ($\Lambda$ is the emission central wavelength and $N_A = 1.45$ is the numerical aperture of the microscope objective). Data of this kind are stable indefinitely upon illumination with laser light of moderate intensity, ~40 W/cm², but under illumination ×50 more intense we observe occasional abrupt loss of signal (Fig. 1c). This may reflect single-step photobleaching[17] but chain scission is another possibility when one considers that the low quantum yield of this optical probe precludes imaging it unless it is stretched. Our final test of single-molecule sensitivity is to illuminate with polarized light. Using polarization-modulated illumination, in Fig. 1d the normalized phosphorescence intensity plotted against

---

[1]Center for Soft and Living Matter, Institute for Basic Science (IBS), Ulsan, South Korea. [2]Department of Chemistry, UNIST, Ulsan, South Korea. ✉e-mail: sgranick@gmail.com

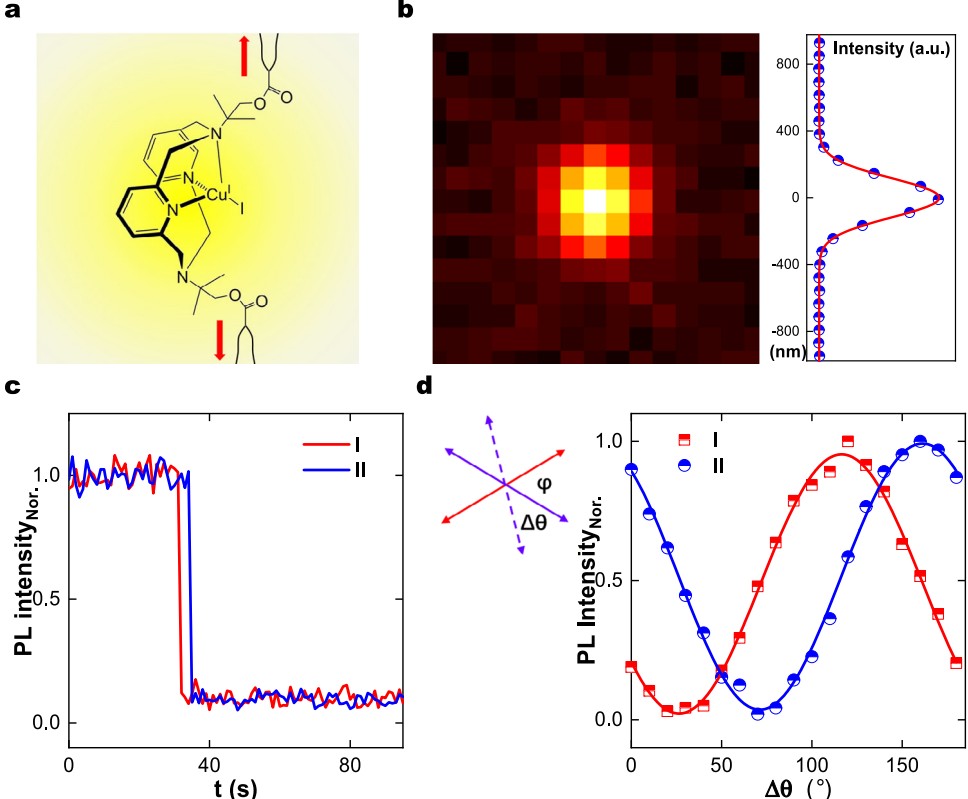

**Fig. 1 | Evidence of single-molecule sensitivity from phosphorescent emission.** **a** Chemical structure of the optical probe covalently attached to polymer strands. Mechanical force (red arrows) promotes brighter emission. Measurements in this figure are in a polymethacrylate (PMA) network stretched to $\lambda = 4$. **b** Optical image with ratio ~1:1000 of phosphorescent to non-phosphorescent crosslinks shows a Gaussian intensity profile with full-width-at-half maximum FWHM = 280 nm, close to the diffraction limit. **c** Photoluminescence (PL) plotted against time for two molecules (I and II) shows single-step photobleaching under excessively strong illumination, ~2 kW/cm$^2$. **d** PL plotted against polarization rotation angle for two molecules (I and II) after rotating a half-wave plate in the excitation path during excitation by linearly-polarized UV light at 355 nm. In the sketch attached to this panel, red and purple arrows show the absorption dipole and excitation dipole directions respectively, with $\Delta\theta$ rotation angle and $\varphi$ the angle between absorption and excitation dipoles. Solid lines are fits to the data according to the expected dependence $\cos^2(\Delta\theta - \varphi)$ for single molecules.

polarization angle shows excellent agreement with the expected relation $\cos^2(\Delta\theta - \varphi)$[18], where $\Delta\theta$ is the rotation angle and $\varphi$ is the original angle between absorption dipole orientation and excitation dipole direction. From this we calculate visibility, $V = (I_{max} - I_{min})/(I_{max} + I_{min})$[19], where $I_{max}$ and $I_{min}$ are the maximum and the minimum intensities, respectively. For the two molecules in Fig. 1d this gives $V = 0.96$ and $0.94$, indicating single-photon emitters[19]. We did not see evidence of aggregation. Taken together, the multiple tests summarized in Fig. 1 indicate that single molecules were detected, but a limitation is that in undeformed networks we did not observe emission sufficiently bright to show up as single molecules. In order to increase the emission intensity, the sample was prestretched. Illustrating this behavior in real time, Supplementary Movie 1 shows emission from five probes during 100 seconds after the network was prestretched to the elongation ratio of four. The spatial fluctuations in these and related movies at other deformations are analyzed below.

Networks contain inevitable geometrical imperfections such as those sketched schematically in Fig. 2a. For these reasons, the history of polymer science includes from the beginning a heated debate over the extent to which macroscopic deformations are identical (or not) at the nm level[20]. The seminal assumption of identical ("affine") deformation, made in seminal theories of rubberlike elasticity (below, we also comment on modern theories that relax this assumption), is easily tested by tracking the relative positions of extensophore crosslinks. Networks prepared to the dimensions roughly of a rubber band were stretched as illustrated by the photo in Fig. 2a. The positions of 12 crosslinks, each of them at three elongation ratios, are shown in Fig. 2b.

For three pairs of these crosslinks, we calculate separations ($d$) in length and width directions, and compare them with the affine expectations as a function of elongation as shown in Fig. 2c. The non-affine parameter $\zeta$, defined as the ratio of non-affine to affine displacement, $\zeta = [d_{affine} - d_{real}]/d_{affine}$[21], is calculated for three crosslink pairs at elongation ratio 4. Differences are evident, indicating that the actual deformations are somewhat less than affine. Moreover, the non-affine parameter differs from spot to spot, suggesting local heterogeneity. This is interesting to know when one considers that the inexpensive and efficient free radical polymerization method we used to construct these networks is widely used in the industrial production of polymer networks.

We now consider local forces. Widely-accepted intuitive expectations of structural heterogeneity (Fig. 2a)[1] predict that some crosslinks will carry load while others will not because they are located in elastically ineffective dangling ends. Consistent with this, when visualizing stretched networks we find that only ~60% of extensophores are sufficiently bright to resolve their positions down to the diffraction limit. Bulk pholuminescence increased in proportion to sample elongation (Fig. 3a) without change of peak shape (Supplementary Fig. 2 and Fig. 3a inset). We approximated the average local force on each crosslink as the macroscopic force to stretch the entire sample, normalized by the average crosslink density per unit area. This is $\bar{F} = \sigma \cdot l^2$, where $\sigma$ is the true stress and $l$ is the average crosslink spacing, estimated from ideal gas equation of entropic elasticity, $G \sim \rho RT$[20] where $G$ is shear modulus (Supplementary Fig. 3), $\rho$ crosslink molar density, and $R$ the gas constant. This implies $N \sim 300$ repeat

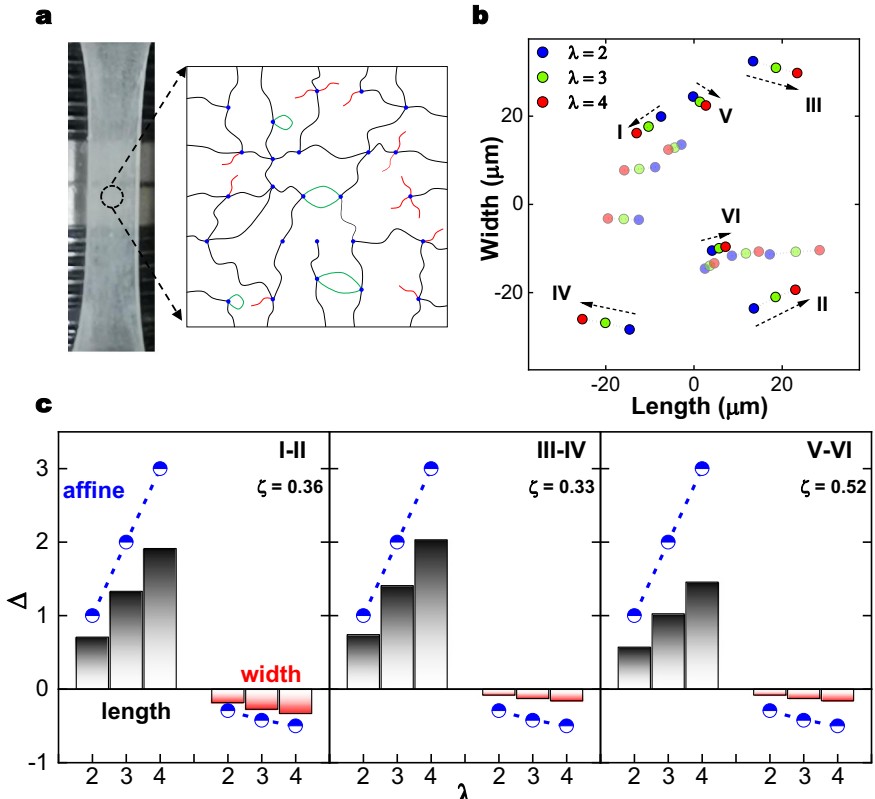

**Fig. 2 | Comparing macroscopic and local deformations. a** A representative photo of a network stretched to $\lambda = 4$, accompanied by a sketch of possible local imperfections of network structure, with crosslinks, polymer strands, and dangling chains/loops indicated by blue, black, red, and green colors, respectively. **b** At three elongation ratios, positions in the length and width directions are mapped for 6 emissive crosslinks (I–VI). **c** At three elongation ratios, separation $\Delta = [\alpha(\lambda) - \alpha_0]/\alpha_0$ in the microscope image between pairs of emissive crosslinks is plotted against the macroscopic stretch, $\lambda$. With comparison to expectations if deformation were affine, the data reveal non-affine deformation quantified as $\zeta$ defined in the text.

units between crosslinks, equivalently the spacing ~8 nm for an ideal random walk. We consider random walk statistics to be a reasonable assumption because the sol fraction was small (see below) and the network was formed in the absence of solvent.

The ideal gas equation $G = \rho RT$ predicts from the known density of chemical crosslinkers the chemically crosslinked shear modulus ($G_x$) about 16 times smaller than measured $G$. This is reasonable physically because our network was lightly crosslinked. If trapped entanglements[22] are the remaining contribution to the measured elastic modulus, then ~50% of the entanglement network $G_e$ is trapped based on the classical papers of Fetters in which the entanglement modulus was characterized to be 0.25 MPa[23]. Admittedly, the ideal gas equation is an approximation. Taking instead a pioneering modern theory of non-affine deforion, the theory of Rubinstein and Panyuokov[24], we estimate $G_x$ and $G_e$ from their Eq. 60: $\frac{\sigma_{\text{eng}}}{\lambda - \lambda^{-2}} = G_x + \frac{G_e}{0.74\lambda + 0.61\lambda^{-0.5} - 0.35}$. From fit to our experimental stress-strain curve (Supplementary Fig. 3), we obtain $G_x = 0.04$ MPa and $G_e = 0.31$ MPa, a result that agrees with the ideal gas theory in the sense that the estimated $G_x$ is about 8 times smaller than $G_e$, consistent with that this network was lightly crosslinked. Therefore, the estimate of this contribution is essentially the same regardless of the theory used.

The respective contribution to $\rho$ of covalent crosslinks and trapped entanglements cannot be assessed more directly from these experiments. Regardless, the estimated force of roughly tens of pN is reassuringly consistent with values obtained independently from directly stretching linear chains using atomic force microscopy (AFM) and optical tweezers[25]. Variability between extensophores suggests the existence of different mechanical microenvironments. The different proportionality between intensity and estimated local force at two

locations within the same network, illustrated in Fig. 3b, was reversible in cyclic deformation (Fig. 3c) and Supplementary Fig. 4, indicating no mechanical failure during extension within the limits of experimental resolution. This approach allows one, after normalizing emission of individual extensophores by the macroscopic average, to construct the map of deformation-dependent position and associated local force illustrated in Fig. 3d. Maps of this kind help to visualize local structure.

Local elasticity is also manifested in thermal fluctuations of crosslink position, as shown in Fig. 4a. Tracking the thermal fluctuations of bright molecules, the exposure time is 0.1 s. The amplitudes of up to ~90 nm are consistent with interpreting that crosslink fluctuations were limited by the mean separation ~130 nm expected from the density of chemical crosslinks. Raw data in Fig. 4b illustrate that fluctuations in the length (elongation) and width directions both appear to be Gaussian with more stiffness in the direction of elongation. Fluctuation amplitude in the elongation and width directions decay in inverse proportion to elongation ($\sim \lambda^{-1}$) and as the inverse square root ($\sim \lambda^{-1/2}$), respectively (Fig. 4c, left top panel). This striking scaling, though not predicted theoretically to the best of our knowledge, may be consistent with the notion of "stretching stiffening" and "tube hardening" of uncrosslinked, entangled polymers[26,27]. In older theory of rubberlike elasticity, large crosslink fluctuations are predicted in a phantom network of chains that unlike entangled chains can cross one another freely[28]. When one makes the assumption that fluctuation amplitude $\beta$ reflects primarily local stiffness with only second-order damping from local viscosity, then it scales as the square root of inverse stiffness, $\beta = \sqrt{\frac{2k_B T}{k}}$[29], where $k$ is elasticity and $k_B$ is the Boltzmann constant. Data analyzed this way extrapolate to identity between $\beta_{//}$ and $\beta_{\perp}$ at $\lambda = 1$ (Fig. 4c, left top panel), indicating that without

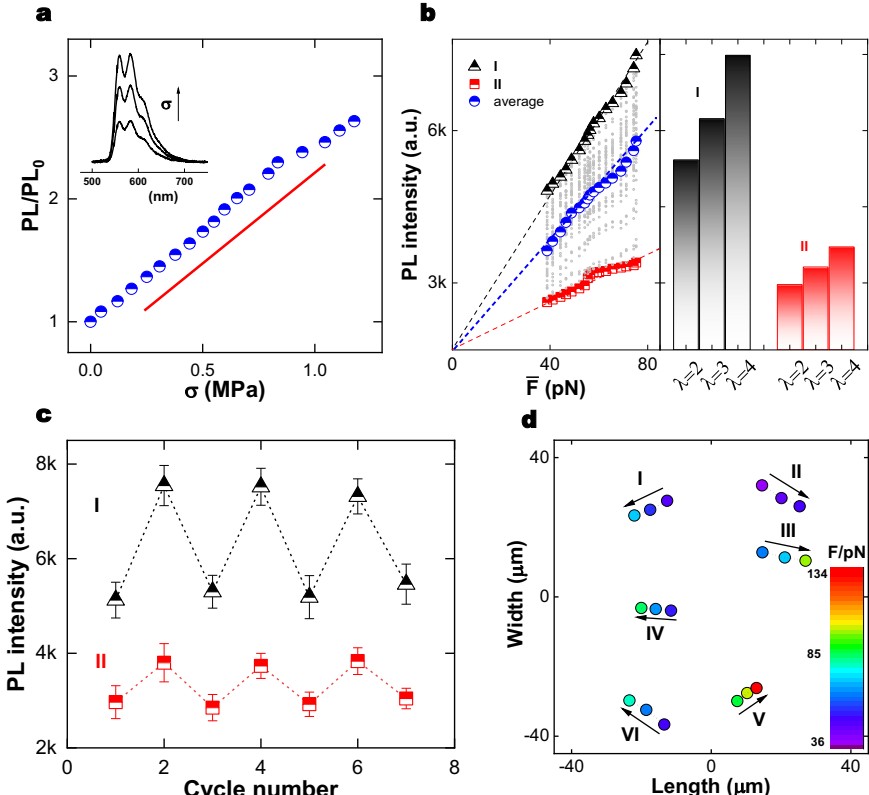

**Fig. 3 | Force within individual crosslinks. a** Photoluminescence (PL) of the macroscopic sample is proportional to stress on the sample. Inset: the PL spectrum shows increased intensity but the same peak shape with increasing stress. **b** PL intensity plotted against average estimated force on each crosslink, for two individual crosslinks (I and II) and for the average of 50 crosslinks. This is accompanied by PL histograms (I and II) at three indicated elongation ratios. **c** PL intensity plotted against cycle number for the same two crosslinks highlighted in panel **b**, demonstrating reversibility in repeated cycles of elongation and subsequent contraction. **d** For six crosslinks (I–VI), the local estimated force is color-coded and spatial positions in the length-width plane are mapped against elongation. The color bar shows the local force, with arrows showing the direction of increasing elongation.

deformation the local structure is nearly isotropic, which pleasingly agrees with physical intuition. The ratio of stiffness in the elongation and width directions increases linearly with elongation (Fig. 4c, left bottom panel). However, histograms associated with individual crosslinks widen with increasing deformation, (Fig. 4c, right panels), suggesting that the mechanical microenvironment became increasingly heterogeneous.

## Discussion

Having established that crosslinks in this polymer system extend, fluctuate, and deform with a wide range of molecular individuality, we conclude that the classical concepts of rubberlike elasticity and macroscopic moduli represent averages over nonuniform mechanical microenvironments. For generations, theorists have considered this possibility[20,24] but the paucity of relevant experimental data prevented the refinement of theory and discrimination between different theoretical approaches.

This proof-of-concept to measure nanoscale/piconewton local forces is limited by the fact that the extensophore we employed requires UV excitation, but there appear no fundamental impediments to developing extensophores sensitive to other excitation wavelengths, thus extending applications of this concept to other synthetic polymer systems and even to gels and tissues of bioengineering interest. For example, DNA tension probes based on mechanofluorescence have been developed though not yet made reversible[30]. Another limitation is that our synthesis of polymer networks by free radical polymerization was relatively uncontrolled, but methods are known in the organic

chemistry and polymer chemistry communities to produce many other kinds of gels[31] in which the extensophore concept can reasonably be applied. In such future studies, it will be interesting to introduce time as a variable and accordingly to use these optical probes to interrogate viscoelastic stress relaxation, provided that it is sufficiently slow. To detect the weak signals from individual extensophore molecules, the needed data acquisition times of 100 ms to sec are comparable to the inverse frequencies and time steps accessible using common rheometers (mechanical testing machines). This will make possible further comparison between classical measurements of bulk mechanical properties and those introduced here to map the diversity of mechanical microenvironments.

## Methods
### Network preparation
Networks are prepared by UV-induced (365 nm) free radical polymerization of methyl acrylate using benzoin as photoinitiator, 1,3-butadiene as non-phosphorescent crosslinker, and phosphorescent crosslinker. The total crosslink concentration is typically 0.8 μM. The molar ratio between phosphorescent and non-phosphorescent is ~1:1000. Unreacted agents are extracted by dialysis in THF solvent (100 mL) exchanged 24 times over 72 h. The sol fraction, measured by weight after dialysis performed in this way, was ~3.5% for these conditions of light crosslinking.

Cast into a Teflon mold, their thickness ~12 μm is optimized to be sufficiently thick to produce free-standing films but sufficiently thin to minimize background light emission that might interfere with single-

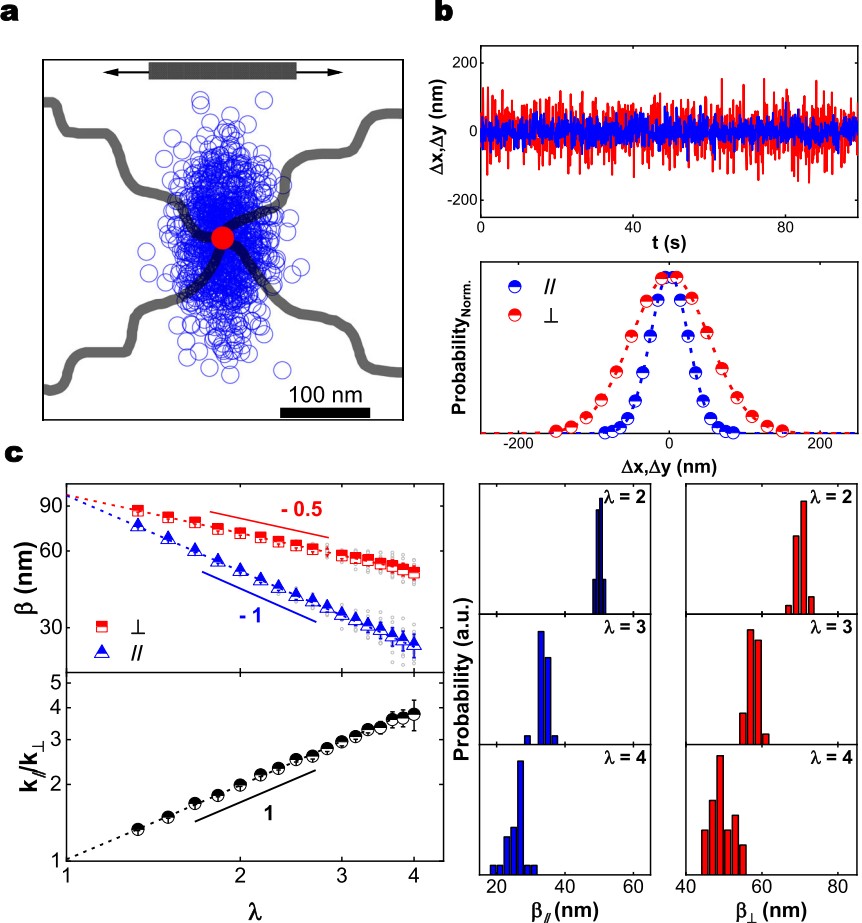

**Fig. 4 | Fluctuation amplitudes of individual crosslinks in elongated networks.**
The dataset is 26 phosphorescent crosslinks selected from those that are brightest.
**a** Map of anisotropic crosslink fluctuations during $100\,s$ at $\lambda = 4$ show less fluctuation in the stretch direction. A superposed schematic representation of the tetra-functional crosslink is included. Black arrows show the elongation direction. **b** Top: representative time trace of position fluctuations in the length ($\parallel$) and width ($\perp$) directions at extension ratio $\lambda = 4$. Bottom: the respective probability distributions are consistent with being Gaussian. **c** Left top: On log-log scales, the fluctuation amplitude $\beta$ obtained from Gaussian fits is plotted against extension ratio, showing scaling with powers $-0.5$ and $-1$ for the width ($\perp$) and length ($\parallel$) directions, respectively. Gray symbols show data for individual molecules. Left bottom: implied local stiffness anisotropy, the stiffness ratio in the elongation and width directions, is plotted against extension ratio. Right: histograms show the respective contributions to crosslink fluctuations of individual phosphorescent crosslinks in length ($\parallel$) and width ($\perp$) directions.

molecule detection. The sample geometry resembles a square rubber band, 6 mm long and 3 mm wide before stretch, but 3 mm of length are dedicated to clamp the sample at both ends for stretch.

To achieve single-molecule optical imaging, the phosphorescent probe concertation should be <1 nM so that the optical probes would be separated by distances large enough to exceed diffraction limitations. This 1 nM concentration achieves micrometer separations. The minimum concentration is set only by the desirability of having several emitter molecules within the microscope's field of view.

## Stretch procedure

Stretch is accomplished using a commercial tensile stage (Deben Mtest200N, UK) using a 200N load cell with 0.02% force precision and 10 mm working distance. The speed of stretch from one elongation to the next is 0.5 mm/min, amounting to strain rate $3 \times 10^{-3}\,s^{-1}$, less than the inverse Rouse relaxation of polymer strands between covalent crosslinks.

## Optical microscopy

The tensile stage sits on an optical microscope. We illuminate (355 nm laser) the sample from the bottom and likewise detect phosphores-cence from the bottom. For single-molecule imaging the crosslinks are diluted (phosphorescent: non-phosphorescent ~1:1000). For calibration of macroscopic force-extension relations, every crosslink was phosphorescent.

Circularly polarized light is used for best excitation efficiency, except for experiments in which we use linearly-polarized light to monitor the polarization dependence. The light emitted is recorded using an EMCCD camera (DU-897U, Andor) with a large pixel density, $512 \times 512$, and a $\times100$ oil objective (Olympus). The field of view is 80 by 80 μm, pixel size 160 nm, and depth of focus 1.5 μm. Video images are analyzed using 2D Gaussian feathering to remove background, using IDL codes written in-house, to give the absolute intensity and positions of the optical probes. To measure the phosphorescence spectrum, the tensile stage is fiber-coupled to an Ocean Optics spectrometer. Pho-toluminescence (PL) is quantified by integrating emission between wavelengths 500 and 700 nm.

## Data statistics

Using 20 freshly prepared samples, we imaged the optical probes at the 17 larger extension ratios specified in Supplementary Fig. 2, repeating each sequence of extensions five times. From these datasets, those subsets were selected in which the same optical probes remained in the focus plane and were visible at every extension ratio.

Data were rejected for which molecules became defocused, as defocus interferes with position localization and quantification of molecular brightness. With these criteria to select viable datasets, we analyzed in total 50 molecules regarding local force and 26 brightest molecules regarding crosslink fluctuations.

## Data availability

The data that underly the plots within this paper and other findings of this study are available from the corresponding author on reasonable request.

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

## Acknowledgements

For discussion, we thank Georgy A. Filonenko and Lingxiang Jiang. This work was supported by the taxpayers of South Korea through the Institute for Basic Science, project code IBS-R020-D1.

## Author contributions

S.G. proposed the project. K.Z. performed the experiments and evaluated data. Y.Z. synthesized the optical probe. B.L. contributed to image analysis. K.Z. and S.G. wrote the manuscript.

## Competing interests

The authors declare no competing interests.
