## [Peer Review File · Nature Communications]

Phosphorescent extensophores expose elastic nonuniformity in polymer networksReviewers' Comments:

Reviewer #1:

Remarks to the Author:

In this manuscript, Granick et. al. used single molecule phosphorescent probes to track nondestructive forces, local deformations and thermal fluctuations in polymer networks. The use of pairs of these single emitters to track their real displacement and compare it to the affine displacement as function of elongation of the macroscopic gel is novel approach and helpful to analyze network properties under external forces.

In my view, this is an amazing work. The presented method links macroscopic forces to forces applied to small molecule mechanophores. In addition, this method can easily be applied to different network topologies with a more regular topology that allow a more homogeneous transduction of macroscopic forces to single molecule crosslinks. Therefore, I support the publication of this work in Nature Comm. However, some comments need to be addressed or corrected:

Minor comments:

- The authors should include raw microscopy images/videos of the ensemble of emitters they are tracking during elongation of the hydrogel
- On page 4 line 5 the text should refer to Figure 2C not 3C

Reviewer #2:

Remarks to the Author:

The authors report on the successful incorporation of a mechanophore capable of 1) reversible operation upon unloading and 2) continuous signal generation in response to loading. They have incorporated the mechanophore into a rubbery network to demonstrate both capabilities. (Sidenote: the sample is referred to several times as a gel, but in fact the sample is dried under vacuum before testing. It is actually just a polymer network, not a gel and the title should be changed to reflect this.) Given extensive recent interest in understanding the contribution of microstructural architecture and its associated local mechanical function to constitutive and damage response in networked materials, I would anticipate that researchers will be eager to incorporate this new technique to address many subtle, yet unresolved problems. Thus, it may be both impactful and of long-lasting use. As written, the work supports the primary claims, e.g., reversible operation and a continuous, force-dependent signal. Only one category of issue should be addressed (see required revision), but it can be done without requiring further experimental effort. The methodology appears sound and sufficient detail is given between the methods and the supplementary information to reproduce the work. I recommend the authors address the following major revisions.

Required Revisions: The authors take the excellent and entirely necessary step of comparing the observations they make about crosslink motion and fluctuation to network elasticity theories. However, it is also apparent that this is not their main area of expertise. This is not meant to be a severe critique but rather a strong suggestion to include an expert on the author list to quickly correct and more impactfully contextualize their findings. I will include a list of examples, but these are not exhaustive, and I would point you toward someone well-versed in this literature to fully address most of these issues: e.g., Michael Rubenstein, Stephan Craig, Jeremiah Johnson. Some potential issues I noticed are:

- "The seminal assumption of identical ("affine") deformation, made in nearly all theories of rubberlike elasticity" – This is simply untrue. Yes, there are several famous affine network theories, but currently there are many more that are not affine. [e.g., Rubinstein & Panyukov, *Macromolecules*, 1997. Davidson & Goulbourne, *JMPS*, 2013. Zhong, et al., *Science*, 2016.]
- Estimation of average local force on each cross-link (p 5): $G = \rho RT$ arises from the affine network model. Perhaps just the same scaling, \sim should be used here since this is forming part of an order-of-magnitude calculation. Inclusion of " $\sim 8\text{nm}$ for random walk" is probably fine given that the sample is in fact in the melt state and not a swollen gel (as "The gel was lightly crosslinked.." included directly

after suggests.) However, this is again approximate as the network was not made in a melt state and so random walk statistics may not apply. More numbers are required here to follow the entire argument to the estimated 10's of pN force.

- "G about 16 times smaller than observed": Some point got lost here. G was measured, used to estimate crosslink density, then used to predict a smaller G? If G_e is to be estimated, the best way to do this is following Rubinstein & Panyukov's method.
- (Extended Data Figure 2). The stress stretch response obtained (engineering stress versus stretch?) is extremely linear to such large stretch values. Does the sample break at $\lambda = 4$? Brittle fracture? Typically a neo-hookean response can be fit to at least a stretch of 1.3-1.4. Can you comment on why the response is so linear to such a large value of stretch?

Small error:

- The legend in Fig. 2C appears to be incorrect. Perhaps "the separation $\Delta = [\]/\alpha_0$ between pairs.... Is plotted against the macroscopic stretch, λ ." ?

Reviewer #3:

Remarks to the Author:

Probing and monitoring molecular level responses upon macroscopic deformation of crosslinked polymer networks is a great challenge. This article reports a technique that allows to monitor these processes by measuring phosphorescence by probes that have been incorporated in a crosslinked polymer network at the single molecule level.

The method reported here very heavily builds upon an earlier report by Filonenko and Khusnutdinova (cited as ref 6 in this article). These two authors in an earlier report describe the use of this probe as incorporated in polyurethane polymers, and demonstrated that the phosphorescence can be used to map stress distribution. The current paper is a further development of this approach; the main novelty seems to be the extension to the observation of single molecule level events, and correlating this with network topology, which is an interesting and powerful complement.

Below are some suggestions that may help to further improve this report:

I would suggest, and believe it is important, that authors better emphasize in the first paragraph of the paper what the novelty of the study is, and what differentiates the method reported here as compared to the earlier paper by Filonenko and Khusnutdinova (see also the comment above)?

In the very first section, I would suggest the authors also to be more explicit about the chemistry used to make the gel, and explicitly state that as a first proof of concept polymethacrylate gels are studied. The type of network prepared is now only mentioned first in the paragraph "evidence of single molecule sensitivity".

Can the authors provide data on the sol and gel fractions of their gels ?

I do not believe this is explicitly mentioned in the paper, but how important / critical it is to control the quantity of probe included in the networks? Have the authors varied this parameter, and is there a minimum amount that must be included, and an upper limit that cannot be exceeded?

The paper states that the extensophore was used to probe force before failure in the network. Do the authors have proof, or would it be possible to do experiments to exclude this ? The data in Figure 1, for example have been obtained on a stretched gel. What do these data look like for a gel in the relaxed state ? Could one differentiate something that is referred to in this paper as photobleaching from scission of the probe molecule (in which case it would act as a chromomechanophore) ?

“Phosphorescent extensophores expose elastic nonuniformity in polymer networks”

All three referees supported publication but with suggestions how to improve the manuscript. In the revised manuscript, we followed the reviewers’ suggestions as described below.

Response to Referee 1

In this manuscript, Granick et. al. used single molecule phosphorescent probes to track nondestructive forces, local deformations and thermal fluctuations in polymer networks. The use of pairs of these single emitters to track their real displacement and compare it to the affine displacement as function of elongation of the macroscopic gel is novel approach and helpful to analyze network properties under external forces. In my view, this is an amazing work. The presented method links macroscopic forces to forces applied to small molecule mechanophores. In addition, this method can easily be applied to different network topologies with a more regular topology that allow a more homogeneous transduction of macroscopic forces to single molecule crosslinks. Therefore, I support the publication of this work in Nature Comm. However, some comments need to be addressed or corrected:

The authors should include raw microscopy images/videos of the ensemble of emitters they are tracking during elongation of the hydrogel

Done. The revised manuscript contains the new Movie S1, showing 5 single-molecule emitters at $\lambda=4$.

On page 4 line 5 the text should refer to Figure 2C not 3C

The typo is fixed. Sorry about that.

Response to Referee 2

The authors report on the successful incorporation of a mechanophore capable of 1) reversible operation upon unloading and 2) continuous signal generation in response to loading. They have incorporated the mechanophore into a rubbery network to demonstrate both capabilities. (Sidenote: the sample is referred to several times as a gel, but in fact the sample is dried under vacuum before testing. It is actually just a polymer network, not a gel and the title should be changed to reflect this.) Given extensive recent interest in understanding the contribution of microstructural architecture and its associated local mechanical function to constitutive and damage response in networked materials, I would anticipate that researchers will be eager to incorporate this new technique to address many subtle, yet unresolved problems. Thus, it may be both impactful and of long-lasting use. As written, the work supports the primary claims, e.g., reversible operation and a continuous, force-dependent signal. Only one category of issue should be addressed (see required revision), but it can be done without requiring further experimental effort. The methodology appears sound and sufficient detail is given between the methods and the supplementary information to reproduce the work. I recommend the authors address the following major revisions

As recommended, the title of the revised manuscript replaces “gel” by “network.”

The authors take the excellent and entirely necessary step of comparing the observations they make about crosslink motion and fluctuation to network elasticity theories. However, it is also apparent that this is not their main area of expertise. This is not meant to be a severe critique but rather a strong suggestion to include an expert on the author list to quickly correct and more impactfully contextualize their findings.

The revised manuscript does a better job of doing justice to the large theoretical literature on this subject. While understanding the referee’s wish to see more knowledgeable discussion of theory than we are able to provide, respectfully we disagree about the referee’s request that we add a theory co-author. To do so would be outside the scope of this paper, but we agree that it would be an excellent idea regarding a future review paper. In this revised manuscript, we do call attention to the recent theories about non-affine behavior.

I will include a list of examples, but these are not exhaustive, and I would point you toward someone well-versed in this literature to fully address most of these issues: e.g., Michael Rubenstein, Stephan Craig, Jeremiah Johnson. Some potential issues I noticed are:

*“The seminal assumption of identical (“affine”) deformation, made in nearly all theories of rubberlike elasticity” – This is simply untrue. Yes, there are several famous affine network theories, but currently there are many more that are not affine. [e.g., Rubinstein & Panyukov, *Macromolecules*, 1997. Davidson & Goulbourne, *JMPS*, 2013. Zhong, et al., *Science*, 2016.]*

The revised manuscript clarifies that while it is true that several famous theories make the affine deformation assumption, many theories do not make this assumption. Thanks for correcting our mistake.

Estimation of average local force on each cross-link (p 5): $G = \rho RT$ arises from the affine network model. Perhaps just the same scaling, \sim should be used here since this is forming part of an order-of-magnitude calculation. Inclusion of “ $\sim 8\text{nm}$ for random walk” is probably fine given that the sample is in fact in the melt state and not a swollen gel (as “The gel was lightly crosslinked..” included directly after suggests.) However, this is again approximate as the network was not made in a melt state and so random walk statistics may not apply. More numbers are required here to follow the entire argument to the estimated 10 's of pN force.

Following the referee’s suggestions:

- 1) we call attention in the revised manuscript to the fact that the ideal gas relation ($G = \rho RT$) is an idealization. More constructively, we now compare ρ estimated from the ideal gas relation to the value obtained using the Rubinstein and Panyukov theory.**
- 2) we write “The gel was lightly crosslinked.”**

- 3) we clarify the reason we consider gelation to have taken place in the melt state: the reason is that it was performed in the absence of solvent. We do recognize that the referee may be thinking that unreacted monomers play the role of being a solvent, but for us (as experimentalists) to evaluate this subtle theoretical hypothesis, which to the best of our understanding is not accepted by theorists universally, would go beyond the scope of this paper. Seeking to respond constructively to the referee's comment, we now do a better job of clarifying that the sol weight fraction of the extracted gel was $\approx 3.5\%$.

"G about 16 times smaller than observed": Some point got lost here. G was measured, used to estimate crosslink density, then used to predict a smaller G? If G_e is to be estimated, the best way to do this is following Rubinstein & Panyukov's method.

The revised manuscript includes comparison the slip-tube model of Rubinstein and Panyukov. As we now do a better job of explaining, this estimate gives almost the same estimate of the contribution from trapped entanglements. Therefore, the estimate of this contribution is essentially the same regardless of the theory used. Thanks for helping us to clarify this point.

(Extended Data Figure 2). The stress stretch response obtained (engineering stress versus stretch?) is extremely linear to such large stretch values. Does the sample break at $\lambda = 4$? Brittle fracture? Typically a neo-hookean response can be fit to at least a stretch of 1.3-1.4. Can you comment on why the response is so linear to such a large value of stretch?

The revised manuscript presents, in Extended Data Figure 2, both ways to calculate stress: (a) the true stress-strain curve, and (b) the engineering stress-strain curve. Our original manuscript had presented only the former.

As we now do a better job of explaining, because the scope of this study was restricted to the regime before mechanical failure, we lack the information that would be needed to answer the referee's questions. The physics of network failure was, by design, outside the scope of our study.

The referee asks why the response is so linear up to a large value of stretch. Speculatively: our strong efforts to measure within regimes of reversible response may contribute. The speed of increasing stretch, from one elongation to the next, amounted to a strain rate $3 \times 10^{-3} \text{ s}^{-1}$, which is less than the inverse Rouse relaxation of polymer strands between crosslinks. This interesting theoretical issue is beyond the scope of this study.

Small error:

- The legend in Fig. 2C appears to be incorrect. Perhaps "the separation $\Delta = [l]/\alpha_0$ between pairs.... Is plotted against the macroscopic stretch, λ ." ?*

The revised manuscript fixes the previously-misleading language.

Response to Referee 3

Probing and monitoring molecular level responses upon macroscopic deformation of crosslinked polymer networks is a great challenge. This article reports a technique that allows to monitor these processes by measuring phosphorescence by probes that have been incorporated in a crosslinked polymer network at the single molecule level.

The method reported here very heavily builds upon an earlier report by Filonenko and Khusnutdinova (cited as ref 6 in this article). These two authors in an earlier report describe the use of this probe as incorporated in polyurethane polymers, and demonstrated that the phosphorescence can be used to map stress distribution. The current paper is a further development of this approach; the main novelty seems to be the extension to the observation of single molecule level events, and correlating this with network topology, which is an interesting and powerful complement.

Below are some suggestions that may help to further improve this report:

I would suggest, and believe it is important, that authors better emphasize in the first paragraph of the paper what the novelty of the study is, and what differentiates the method reported here as compared to the earlier paper by Filonenko and Khusnutdinova (see also the comment above)?

The revised manuscript does a better job of discriminating the novelty of this study and the important earlier study by Filonenko and Khusnutdinova. As the referee recommends, this is presented in the first paragraph. Briefly: Filonenko and Khusnutdinova pioneered these molecules but used them only to characterize macroscopic, ensemble-averaged deformations. To use them for single-molecule study is unique to the present study. We also do a better job of clarifying that in order to enable covalent attachment of these optical probes to polymer networks, we implemented a small modification of the synthesis used by Filonenko and Khusnutdinova.

In the very first section, I would suggest the authors also to be more explicit about the chemistry used to make the gel, and explicitly state that as a first proof of concept polymethacrylate gels are studied. The type of network prepared is now only mentioned first in the paragraph “evidence of single molecule sensitivity”.

Done. The revised manuscript explains more clearly that the polymer network was made using UV-induced free radical polymerization and that as proof of concept, we crosslinked polymethacrylate (PMA). This we now do in the first paragraph.

Can the authors provide data on the sol and gel fractions of their gels ?

Done. We now do a better job of explaining that the networks were measured after removing the sol fraction, $\approx 3.5\%$. We also clarify that before experiments, the sol was removed by dialysis and then the network was dried in vacuum.

I do not believe this is explicitly mentioned in the paper, but how important / critical it is to control the quantity of probe included in the networks? Have the authors varied

this parameter, and is there a minimum amount that must be included, and an upper limit that cannot be exceeded?

Done. The revised manuscript does a better job of explaining (previously, this important point was buried in the text) that to achieve single-molecule optical imaging, the phosphorescent probe concentration should be < 1 nM so that the optical probes would be separated by distances large enough to exceed diffraction limitations. The 1 nM concentration achieves micrometer separations. The minimum concentration is set only by the desirability of having several emitter molecules within the microscope's field of view.

The paper states that the extensophore was used to probe force before failure in the network. Do the authors have proof, or would it be possible to do experiments to exclude this? The data in Figure 1, for example have been obtained on a stretched gel. What do these data look like for a gel in the relaxed state? Could one differentiate something that is referred to in this paper as photobleaching from scission of the probe molecule (in which case it would act as a chromomechanophore)?

Following up on the referee's questions:

- 1. We consider reversibility of our measurements to signify that failure, if it contributes, was a second-order contribution below the experimental resolution. This is noted in the revised manuscript.**
- 2. Though low quantum yield of the probe precluded measurements in the relaxed state, from extrapolation to the relaxed state (Fig. 3B) one can infer the (extremely small) forces applied to the probe in the relaxed state. This is noted in the revised manuscript.**
- 3. As the referee pointed out, abrupt loss of emission intensity might originate in chain scission, not in the photobleaching we had presumed. This is noted in the revised manuscript.**

Reviewers' Comments:

Reviewer #1:

Remarks to the Author:

The authors have addressed all comments and the MS is in shape for acceptance

Reviewer #2:

Remarks to the Author:

The authors have sufficiently addressed all my concerns. This is an excellent manuscript I expect to be highly impactful.

Reviewer #3:

Remarks to the Author:

I am pleased with the responses that have been provided by the authors and with the changes that have been made to the manuscript, and support publication of this work.

“Phosphorescent extensophores expose elastic nonuniformity in polymer networks”

All three referees supported publication without further modification.